Study of TRAF3IP3 for prognosis and immune infiltration in hepatocellular carcinoma

Wang Xing 1
Gao Xin 1
Liu Airu 1
Qin Yan 2
Ni Zhi-Yu 3 nizhiyu@hbu.edu.cn
Zhang Xiao Lan 1 xiaolanzh@126.com
1 Hebei Key Laboratory of Gastroenterology, Hebei Institute of Gastroenterology, Hebei Clinical Research Center for Digestive Diseases, Department of Gastroenterology, The Second Hospital of Hebei Medical University , Shijiazhuang , China
2 Central Laboratory, Affiliated Hospital of Hebei University, Hebei Collaborative Innovation Center of Tumor Microecological Metabolism Regulation, Hebei Key Laboratory of Precise Imaging of Inflammation Related Tumors, Hebei University , Baoding , China
3 The Affiliated Hospital of Hebei University, School of Basic Medical Science, Hebei University , Baoding , China
Oppert Brenda
Electronic publication date: 2024 Dec 12
Publication date: 2024
Volume: 12
Electronic Location ID: e18538
Received 2023 Oct 30; Accepted 2024 Oct 25
Copyright: © 2024 Wang et al.
Copyright year: 2024
Copyright holder: Wang et al.
License: This is an open access article distributed under the terms of the Creative Commons Attribution License, which permits unrestricted use, distribution, reproduction and adaptation in any medium and for any purpose provided that it is properly attributed. For attribution, the original author(s), title, publication source (PeerJ) and either DOI or URL of the article must be cited.
License URL: https://creativecommons.org/licenses/by/4.0/

Keywords: TRAF3IP3, Lymphocyte, Immune cell infiltration, Hepatocellular carcinoma, Immunotherapy, CD8+ T cells, Biomarker, Prognosis, CTLA-4, PD-1

Funding: The authors received no funding for this work.

==============================
Background

Tumor necrosis factor receptor-associated factor 3 (TRAF3)-interacting protein 3 (TRAF3IP3) expressed in various tumor cell. However, its role in hepatocellular carcinoma (HCC) was unclear. We aimed to demonstrate the relationship between TRAF3IP3 and HCC and explore the potential role of TRAF3IP3 in HCC.

Methods

The Cancer Genome Atlas (TCGA), Gene Expression Omnibus (GEO), Genotype-Tissue Expression (GTEx), KM-Plotter, University of Alabama at Birmingham Cancer data analysis Portal (UALCAN), and Xiantao Academic Online Website were utilized for the systematic analysis of TRAF3IP3. This analysis included mRNA expression, protein expression, prognostic value, enrichment analysis, and immune cell infiltration in HCC. Subsequently, immunohistochemistry was performed to assess the expression levels of TRAF3IP3 in both cancer and non-cancer tissues of patients with HCC.

Results

Analysis of public databases and immunohistochemical staining on 20 pairs of samples confirmed a decrease in TRAF3IP3 expression in HCC. Both the TCGA database and GSE14520 indicated that patients with high TRAF3IP3 expression had a more favorable prognosis in terms of overall survival (OS) and progression-free interval (PFI), as shown by KM curve results. Multivariate Cox regression analysis further demonstrated that high TRAF3IP3 expression was an independent protective factor for HCC prognosis (hazard ratio (HR): 0.619, 95% confidence interval (CI) [0.399–0.959]; p < 0.05). In the high TRAF3IP3 expression group, various immune response-related molecular pathways, particularly B lymphocyte-mediated pathways, were activated. The level of TRAF3IP3 expression showed a significant correlation with the presence of tumor-infiltrating CD8+ T cells. Additionally, a positive correlation was observed between immunophenoscore (IPS) and TRAF3IP3 expression. Notably, the half-maximal inhibitory concentration (IC50) of commonly used chemotherapeutic drugs, such as lapatinib and mitomycin, was inversely associated with TRAF3IP3 expression in HCC patients.

Conclusion

TRAF3IP3 may be as a novel and promising biomarker for prognosis prediction and immunological evaluation of HCC.

Introduction

Hepatocellular carcinoma (HCC) is a prevalent and severe cancer, ranking sixth in frequency and accounting for 4.7% of cases according to Global Cancer Statistics 2020. With the third-highest mortality rate among all malignancies, HCC poses a significant public health threat (Sung et al., 2021). The advancement of various therapies such as interventional radiofrequency ablation (Perez-Romasanta et al., 2021), chemotherapy, targeted therapy (Cheon et al., 2022), surgical resection (Gunasekaran et al., 2021), and liver transplantation (Linares et al., 2019) has led to significant progress in HCC treatment. However, the 5-year survival rate for HCC patients in China remains low due to high rates of recurrence and metastasis (Feng et al., 2019; Zeng et al., 2018). The lack of effective biomarkers makes predicting overall survival challenging for this patient population, highlighting the need for new and potent prognostic biomarkers in HCC.

Tumor necrosis factor receptor-associated factor 3 (TRAF3)-interacting protein 3 (TRAF3IP3), also known as TRAF3-interacting Jun N-terminal kinase-activating modulator (T3JAM), forms a complex with TRAF3 that activates the JNK signaling pathway (Dadgostar et al., 2003). TRAF3IP3 inhibits mTORC1 on lysosomes, thereby limiting glycolytic metabolism and supporting the stability and function of Treg cells (Yu et al., 2018). Through knockout (KO) mouse models, it has been shown that TRAF3IP3 enhances autophagy and boosts the TI-II immune response by interacting with ATG16L1, ultimately promoting the survival of marginal zone (MZ) B cells (Peng et al., 2015). Primarily involved in regulating lymphocyte development, TRAF3IP3 has been linked to tumor progression (Sanati et al., 2018). However, its role in different tumors varies. High expression of TRAF3IP3 in melanoma is associated with increased tumor proliferation, invasion, and metastasis, leading to poor outcomes in melanoma patients (Nasarre et al., 2018). On the other hand, TRAF3IP3 may facilitate glioma development by activating the ERK signaling pathway (Lin et al., 2022; Yang et al., 2021), while acting as a protective gene in patients with lung adenocarcinoma (Liu et al., 2021). In HCC, tumor infiltrating lymphocytes (TILs) play a crucial role in the host’s anticancer response (Hernandez-Gea et al., 2013) and impact disease progression and response to immunotherapy. Despite the significance of TILs in HCC, the role of TRAF3IP3 in this context remains unexplored, highlighting the need for further research to uncover its biological function and prognostic value in HCC.

In this study, we analyzed TRAF3IP3 in HCC using datasets from the Cancer Genome Atlas (TCGA), Genotype-Tissue Expression (GTEx) and various online bioinformatic analysis platforms. We first examined TRAF3IP3 expression levels in HCC and investigated their correlation with HCC prognosis and clinicopathological parameters. Additionally, we explored the relationship between TRAF3IP3 and tumor microenvironment (TME) scores, immune cell infiltration, and immune checkpoint. Furthermore, we revealed the connection between TRAF3IP3 and tumor immunotherapy response. Immunohistochemistry (IHC) was utilized to validate the differential expression of TRAF3IP3 in HCC tissues. Overall, this study suggests the potential of TRAF3IP3 as a predictive biomarker for prognosis and immunotherapy response in HCC, warranting further investigation.

Materials and Methods

Data collection

RNA sequencing data, transcriptome profiles, and clinical information of 531 cases were obtained from the University of California Santa Cruz (UCSC) Xena database (https://xenabrowser.net/datapages/). This dataset included 110 cases of GTEx (http://commonfund.nih.gov/GTEx/) normal data, 50 cases of TCGA (http://cancergenome.nih.gov) para-cancer data, and 371 cases of TCGA tumor data. The normal group comprised both GTEx normal data and TCGA para-cancer data, while the tumor group consisted of TCGA tumor data. To validate the mRNA expression of TRAF3IP3, GSE121248 dataset from GEO database was utilized. Additionally, histograms of TRAF3IP3 protein expression in normal and primary tumor tissues were obtained from the UALCAN portal (https://ualcan.path.uab.edu/analysis-prot.html) using Clinical Proteomic Tumor Analysis Consortium (CPTAC) data. Furthermore, GSE14520 was downloaded for survival analyses and functional enrichment analysis. Immune-cell infiltration and half-maximal inhibitory concentration (IC50) of commonly used chemotherapeutic drugs were separately verified using external cohorts (GSE121248, GSE60502, GSE76427, GSE102451).

Clinical samples

Formalin-fixed paraffin-embedded (FFPE) tissue samples were acquired with approval from the Ethics Committee of the Second Hospital of Hebei Medical University (approval number: 2022-R706). The study enrolled 20 patients diagnosed with HCC who underwent surgical resection at the same hospital in 2021. Inclusion criteria included individuals aged 18–75 years with a confirmed histological diagnosis and no prior neoadjuvant chemotherapy or radiotherapy. Patients with severe complications or other malignancies were excluded (Luo et al., 2022a). Of the 20 HCC patients, 65% (13/20) were male and 35% (7/20) were female, with ages ranging from 22 to 70 years (median age, 53.6 years). Moreover, 20% (4/20) of cases were classified as stage II, while 80% (16/20) were classified as stage III. Among the patients, 70% (14/20) were positive for HBsAg, and 5% (1/20) were positive for HCV. Additionally, 65% (13/20) of cases did not exhibit histologic evidence of cirrhosis, while 35% (7/20) did show cirrhosis. Furthermore, there were nine cases of Alpha-Fetoprotein (AFP) negative and 11 cases of AFP positive (Table 1).

Table 1 Basic clinical information sheet.

Patient	Gender	Age	Pathologic stage	HBV	HCV	Cirrhosis	AFP	
A1	Male	22	Stage III	(+)	(−)	(−)	(+)	
A2	Male	36	Stage III	(+)	(−)	(−)	(+)	
A3	Male	36	Stage III	(+)	(−)	(−)	(−)	
A4	Male	40	Stage II	(+)	(−)	(+)	(−)	
A5	Male	42	Stage II	(+)	(−)	(+)	(+)	
A6	Male	43	Stage III	(+)	(−)	(+)	(−)	
A7	Female	45	Stage III	(+)	(−)	(+)	(+)	
A8	Female	46	Stage III	(−)	(−)	(−)	(+)	
A9	Male	53	Stage III	(+)	(−)	(−)	(−)	
A10	Female	56	Stage III	(+)	(−)	(−)	(−)	
A11	Female	59	Stage III	(−)	(−)	(+)	(−)	
A12	Male	62	Stage III	(−)	(+)	(+)	(+)	
A13	Female	62	Stage II	(+)	(−)	(−)	(−)	
A14	Male	64	Stage III	(+)	(−)	(−)	(+)	
A15	Male	65	Stage III	(−)	(−)	(−)	(−)	
A16	Male	66	Stage III	(−)	(−)	(−)	(−)	
A17	Female	67	Stage III	(+)	(−)	(−)	(+)	
A18	Female	68	Stage III	(−)	(−)	(−)	(+)	
A19	Male	70	Stage II	(+)	(−)	(+)	(+)	
A20	Male	70	Stage III	(+)	(−)	(−)	(+)	

Immunohistochemical staining

Paraffin-embedded tissue sections were sliced into 4-μm sections and stained for immunohistochemistry analysis. The slides were incubated with a 1:25 dilution of primary antibody against rabbit TRAF3IP3 (35969-4926; SAB, Jiangsu, China) at 4 °C overnight and were then incubated with the SP (Rabbit) IHC kit (PV-9001; ZSGB–BIO, Beijing, China) secondary antibody. Diaminobenzidine tetrahydrochloride was used for the chromogenic reactions. The ImageJ software (NIH, Bethesda, MD, USA) was utilized for calculating the cumulative optical density (IOD) and pixel area of tissue. The immunohistochemical results were then represented as the average optical density (AOD).

Survival analysis

The prognostic value of TRAF3IP3 was evaluated using clinical data from the TCGA database and an external cohort (GSE14520, n = 221). Bioinformatics analysis was conducted on the Xiantao Academic Online Website (https://www.xiantao.love/) using the R language. Statistical analysis was performed using “survival” (version 3.3.1) package, with visualization done using “survminer” (version 0.4.9) and “ggplot2” (version 3.3.6) packages. Kaplan–Meier (K–M) survival curves were used to compare differences in overall survival (OS) and progression-free interval (PFI) between high and low TRAF3IP3 expression groups.

Development and validation of nomogram

The patients were classified into high- and low-expression groups based on the median expression of TRAF3IP3. Univariate and multivariate analyses were conducted to assess the independent prognostic significance of TRAF3IP3. Subsequently, a genomic-clinical nomogram was developed using the ‘rms’ R package to predict 1-, 3-, and 5-year OS for patients with HCC. The accuracy of the nomogram was assessed through calibration curves.

Functional enrichment

Patients with HCC were categorized into high- and low-expression groups using the median level of TRAF3IP3 mRNA expression. The differentially expressed genes (DEGs) between these groups (|log FC| ≥ 1 and FDR < 0.05) were identified using the ‘limma’ R package. Spearman correlations between TRAF3IP3 and the top 50 DEGs were evaluated and visualized through a heatmap. These DEGs were then utilized for Gene Ontology (GO) enrichment analyses, focusing on biological pathways (BPs), cellular components (CCs), and molecular functions (MFs). Additionally, Gene Set Enrichment Analysis (GSEA) was utilized to explore the molecular pathways associated with TRAF3IP3 in HCC using the MSigDB database (C5.Go.v7.4.Symbols.GMT) (Subramanian et al., 2005). The top five results were extracted and stored. Statistical analysis and graphical representation were carried out using the clusterProfiler R package (Yu et al., 2012).

Immune evaluation

To investigate the potential relationship between TRAF3IP3 expression and the tumor microenvironment (TME), we used the ‘ESTIMATE’ package (Yoshihara et al., 2013) to calculate the immune score, stromal score, and estimate score of tumor samples base on TCGA and GEO database. Additionally, we gauged the proportions of tumor-infiltrating immune cells in HCC through CIBERSORT algorithm (http://cibersort.stanford.edu/) (Newman et al., 2015) and evaluated the correlations between TRAF3IP3 expression and the immune cell subpopulation. A p-value < 0.05 was set as the criterion to select lymphocytes possibly affected by TRAF3IP3 expression. Moreover, Spearman correlation analysis was utilized in this study to investigate the association between TRAF3IP3 expression and immune checkpoint genes including CD274, CTLA4, HAVCR2, LAG3, PDCD1, PDCD1LG2, TIGIT, as well as chemokines and their receptors in HCC.

Prediction of therapeutic responses

Based on data from HCC patients in TCGA, we assessed the potential sensitivities of TRAF3IP3 in both immunotherapy and targeted/chemotherapies. Immunotherapy responses were evaluated using the IPS, calculated through machine learning and sourced from The Cancer Immunome Atlas (TCIA) (https://tcia.at/home) (Charoentong et al., 2017). Differences in IPS between TRAF3IP3 high- and low-expression groups were compared using the Wilcoxon test, with a higher IPS indicating greater sensitivity to immunotherapy. Additionally, the IC50 values of 138 drugs were estimated using the ‘pRRophetic’ package (Geeleher, Cox & Huang, 2014) and then normalized to assess TRAF3IP3’s predictive capacity for responses to targeted/chemotherapies.

Statistical analysis

The expression differences of TRAF3IP3 were analyzed using a Chi-square test. Cox analysis was conducted to identify independent prognostic risk factors for HCC patients. Survival differences were assessed using Kaplan–Meier curves and the log-rank test. Additionally, Spearman correlation analysis was performed to investigate correlations. Statistical analyses were performed by using GraphPad Prism 7.0. Bioinformatics analysis was completed by R packages 4.1.3 (https://www.r-project.org). A statistical significance level of p < 0.05 was used to assess the results.

Results

Identification of TRAF3IP3 expression in HCC tissues

The study integrated data from TCGA and GTEx to analyze the expression differences of TRAF3IP3 in HCC. Results showed that TRAF3IP3 mRNA expression was lower in HCC tumor tissues compared to normal tissues (Fig. 1A). Analysis of the GSE121248 and GSE14520 dataset further confirmed this decrease in TRAF3IP3 expression in HCC tissues (Fig. 1B; Fig. S1). Protein expression of TRAF3IP3 in HCC was assessed using the ‘CPTAC Analysis’ module of UALCAN, revealing a significant reduction in protein levels in HCC compared to adjacent normal tissues (Fig. 1C). Immunohistochemistry on 20 paired HCC samples validated these findings, showing strong immunohistochemical staining of TRAF3IP3 in the cytoplasm of adjacent noncancerous tissues, while weak staining was observed in HCC cancer tissues (Fig. 1D). The AOD value of TRAF3IP3 immunohistochemical staining in adjacent normal liver tissues was higher compared to tumor tissues, indicating lower TRAF3IP3 expression in tumor tissues than in adjacent normal liver tissues (Fig. 1E).

Figure 1 TRAF3IP3 was downregulate in hepatocellular carcinoma.

(A) TRAF3IP3 expression in HCC tissues and normal tissues from TCGA and GTEx data. (B) The mRNA expression of TRAF3IP3 in HCC tissues and matched adjacent tissues from GSE121248. (C) The protein expression of TRAF3IP3 between normal and HCC tissues from UALCAN. (D) Representative IHC staining of TRAF3IP3 in HCC tissues and adjacent normal tissues. (E) The quantitative analysis of TRAF3IP3 expression in 20 paired HCC tissue based on average optical density (**p < 0.01; ***p < 0.001).

Relationship between TRAF3IP3 mRNA expression and the prognosis of HCC patients

The prognostic significance of TRAF3IP3 in HCC was further evaluated using K–M survival curves for OS and PFI. Higher TRAF3IP3 expression correlated with improved OS at 80 months follow-up (Fig. 2A, p = 0.042) and better PFI (Fig. 2B, p = 0.003) in HCC patients. Additionally, analysis of an external cohort (GSE14520, n = 221) confirmed that increased TRAF3IP3 expression was associated with a favorable prognosis for HCC patients (Fig. 2C, p = 0.011).

Figure 2 The prognostic value of TRAF3IP3 in HCC.

(A) The overall survival (OS) of hepatocellular carcinoma (HCC) patients with high TRAF3IP3 expression was better than that of patients with low TRAF3IP3 expression over an 80-month follow-up period. (B) TRAF3IP3 expression was found to be positively associated with progression-free interval (PFI) in HCC patients. (C) Further analysis of an external cohort from the Gene Expression Omnibus (GEO) confirmed that elevated TRAF3IP3 expression was linked to a favorable survival outcome.

TRAF3IP3 correlates with clinical features and outcomes in HCC

The baseline datasheet showed that TRAF3IP3 was significantly correlated with the tumor status, weight and the fibrosis Ishak score (Table 2). Univariate Cox regression analyses further demonstrated that TRAF3IP3 was associated with a better prognosis (hazard ratio (HR): 0.705, 95% confidence interval (CI) [0.497–0.999]) (Fig. 3A). Moreover, multivariate Cox regression analyses affirmed that low expression of TRAF3IP3 was an independent factor linked to OS (HR: 0.619, 95% CI [0.399–0.959]) (Fig. 3B). Based on the multivariate Cox regression analyses, nomogram prediction models were established (Fig. 3C). We performed calibration analysis on the nomograms to verify the validity of the predictive models (Fig. 3D). These results confirmed TRAF3IP3 as an independent favorable factor for HCC survival.

Table 2 The relevance between TRAF3IP3 expression and clinical characteristics.

Characteristic	Low expression of TRAF3IP3	High expression of TRAF3IP3	p	
n	187	187		
T stage, n (%)			0.990	
T1	91 (24.5%)	92 (24.8%)		
T2	48 (12.9%)	47 (12.7%)		
T3	41 (11.1%)	39 (10.5%)		
T4	7 (1.9%)	6 (1.6%)		
N stage, n (%)			1.000	
N0	133 (51.6%)	121 (46.9%)		
N1	2 (0.8%)	2 (0.8%)		
M stage, n (%)			0.624	
M0	140 (51.5%)	128 (47.1%)		
M1	3 (1.1%)	1 (0.4%)		
Pathologic stage, n (%)			0.683	
Stage I	86 (24.6%)	87 (24.9%)		
Stage II	44 (12.6%)	43 (12.3%)		
Stage III	42 (12%)	43 (12.3%)		
Stage IV	4 (1.1%)	1 (0.3%)		
Tumor status, n (%)			0.041	
Tumor free	89 (25.1%)	113 (31.8%)		
With tumor	85 (23.9%)	68 (19.2%)		
Gender, n (%)			0.658	
Female	58 (15.5%)	63 (16.8%)		
Male	129 (34.5%)	124 (33.2%)		
Race, n (%)			0.571	
Asian	85 (23.5%)	75 (20.7%)		
Black or African American	8 (2.2%)	9 (2.5%)		
White	88 (24.3%)	97 (26.8%)		
Age, n (%)			0.643	
<=60	86 (23.1%)	91 (24.4%)		
>60	101 (27.1%)	95 (25.5%)		
Weight, n (%)			0.041	
<=70	102 (29.5%)	82 (23.7%)		
>70	71 (20.5%)	91 (26.3%)		
Height, n (%)			0.776	
<170	102 (29.9%)	99 (29%)		
>=170	68 (19.9%)	72 (21.1%)		
BMI, n (%)			0.055	
<=25	97 (28.8%)	80 (23.7%)		
>25	70 (20.8%)	90 (26.7%)		
Vascular invasion, n (%)			0.550	
No	109 (34.3%)	99 (31.1%)		
Yes	53 (16.7%)	57 (17.9%)		
Child-Pugh grade, n (%)			0.211	
A	117 (48.5%)	102 (42.3%)		
B	14 (5.8%)	7 (2.9%)		
C	0 (0%)	1 (0.4%)		
Prothrombin time, n (%)			0.305	
<=4	111 (37.4%)	97 (32.7%)		
>4	41 (13.8%)	48 (16.2%)		
Albumin (g/dl), n (%)			0.500	
<3.5	32 (10.7%)	37 (12.3%)		
>=3.5	120 (40%)	111 (37%)		
AFP (ng/ml), n (%)			0.139	
<=400	114 (40.7%)	101 (36.1%)		
>400	27 (9.6%)	38 (13.6%)		
Adjacent hepatic tissue inflammation, n (%)			0.703	
None	58 (24.5%)	60 (25.3%)		
Mild	54 (22.8%)	47 (19.8%)		
Severe	8 (3.4%)	10 (4.2%)		
Histologic grade, n (%)			0.668	
G1	29 (7.9%)	26 (7%)		
G2	87 (23.6%)	91 (24.7%)		
G3	62 (16.8%)	62 (16.8%)		
G4	8 (2.2%)	4 (1.1%)		
Residual tumor, n (%)			0.623	
R0	167 (48.4%)	160 (46.4%)		
R1	10 (2.9%)	7 (2%)		
R2	1 (0.3%)	0 (0%)		
Fibrosis ishak score, n (%)			0.047	
0	47 (21.9%)	28 (13%)		
1/2	11 (5.1%)	20 (9.3%)		
3/4	16 (7.4%)	12 (5.6%)		
5/6	38 (17.7%)	43 (20%)		

Figure 3 TRAF3IP3 correlates with clinical features and outcomes in HCC.

(A and B) A forest plot of univariate and multivariate Cox regression analyses with OS in HCC from TCGA database. (C) Construction of a prognostic nomogram in HCC. (D) Nomogram calibration analysis of HCC prognostic data.

Analysis of functional enrichment

Figure 4A displays the top 50 genes that were both positively and negatively related with TRAF3IP3 in HCC. Subsequently, DEGs underwent Kyoto Encyclopedia of Genes and Genomes (KEGG) and GO enrichment analysis to assess the biological pathways and functions associated with TRAF3IP3. TRAF3IP3-related DEGs were enriched in (i) BPs: positive regulation of lymphocyte activation, lymphocyte mediated immunity; (ii) CCs: T cell receptor; and (iii) MFs: antigen binding, immune receptor activity, immunoglobulin receptor binding (Fig. 4B). In addition, the top 15 KEGG pathways for TRAF3IP3 were shown in Fig. 4C. Among these pathways, many immune-related pathways were highly associated with TRAF3IP3, including cytokine-cytokine receptor interaction, chemokine signaling pathway, Th17 cell differentiation, Th1 and Th2 cell differentiation and intestinal immune network for IgA production in HCC. The molecular pathways linked to TRAF3IP3 in HCC were investigated using the GSEA. Among the GO terms, the top five signaling pathways influenced by TRAF3IP3 were enriched mainly in immune-related activities, including B-cell activation, B-cell mediated immunity, and the B-cell receptor signaling pathway in HCC (Fig. 4D). The results were further validated by an external cohort (GSE174570, n = 43) (Fig. S2).

Figure 4 Analysis of functional enrichment.

(A) Heatmap showing the top 50 genes positively and negatively correlated with TRAF3IP3 in HCC. (B) GO enrichment bubble plot. (C) Ridge plots of GSEA enrichment. (D) Top 15 KEGG enrichment pathways in HCC.

Correlation between TRAF3IP3 expression and immune infiltration

To further investigate the influence of TRAF3IP3 on HCC development, we initially examined its role in the TME, which is composed mainly of tumor cells, immune cells, and stromal cells.. Based on data from the TCGA database, we utilized the ESTIMATE algorithm to calculate the stroma score, immune score, and estimate score of tumor samples and assessed the correlation between TRAF3IP3 expression levels and those scores. As shown in Fig. 5A, the group with high TRAF3IP3 expression had significantly higher stroma score, immune score, and estimate score than the group with low expression (p < 0.001). Validation with external cohorts (GSE121248, n = 107; GSE60502, n = 36; GSE76427, n = 167; GSE102451, n = 59) confirmed these findings (Fig. S3). Immunotherapy is increasingly demonstrating its advantages in tumor treatment, prompting us to analyze the relationship between TRAF3IP3 and immune factors, such as immune cells. In the analysis of tumor-infiltrating immunocyte profiles using the CIBERSORT algorithm with TCGA database, we identified the levels of infiltration of 22 immune cell types. The high-TRAF3IP3 group exhibited elevated levels of activated CD4+ memory T cells and CD8+ T cells, while M0 macrophages displayed lower levels of infiltration (Fig. 5B). Additionally, we investigated the relationship between immune cell infiltration and TRAF3IP3 expression in HCC. Our findings revealed a significant positive correlation between TRAF3IP3 expression and CD8 T cells (R = 0.48, p = 8.9e−06), CD4 memory-activated T cells (R = 0.44, p = 4.2e−05), and follicular helper T cells (R = 0.36, p = 0.00095). Conversely, the infiltration of M0 macrophages (R = 0.27, p = 0.015) and M2 macrophages (R = 0.28, p = 0.013) showed a negative correlation with TRAF3IP3 expression (Figs. 5C and 5D). Our study uncovered notable correlations between the expression of various immune checkpoints and TRAF3IP3 in patients with HCC (Figs. 6A and 6B), suggesting that TRAF3IP3 could serve as a reliable indicator for predicting immune checkpoint expression levels in HCC. Additionally, we found a positive association between TRAF3IP3 expression and chemokines such as CCL5 and CCL9, as well as their receptor CCR5, and their receptor CCR5 (Figs. S4A and S4B).

Figure 5 Immune infiltration analysis of TRAF3IP3.

(A) Comparison of stroma score, immune score, and estimate score between TRAF3IP3 high and low expression groups. (B) Comparison of immune cell infiltration between TRAF3IP3 high and low expression groups. (C and D) Correlation analysis between the expression level of TRAF3IP3 and immune cell infiltration (*p < 0.05; **p < 0.01; ***p < 0.001).

Figure 6 Analysis of the relationship between TRAF3IP3 expression and immune checkpoints in HCC.

(A and B), The relationship between TRAF3IP3 expression and immune checkpoint genes (*p < 0.05; **p < 0.01; ***p < 0.001).

TRAF3IP3 predicts efficacy of chemotherapy and immunotherapy response

The significant correlation between TRAF3IP3 and multiple immune checkpoints piqued our interest, particularly in the context of the promising clinical application of immune checkpoint therapy in cancer treatment. Initially, we investigated the association between TRAF3IP3 and IPS in patients with HCC. IPS was a known predictor of response to immune checkpoint therapy, determined by evaluating key immune-related gene expressions. Our analysis revealed that the IPS score for IPS-CTLA4 blocker and IPS-PD1 blocker was notably higher in the low-expression group, indicating increased sensitivity to immunotherapy (Figs. 7A and 7B). To investigate the relationship between TRAF3IP3 expression levels and the prognosis of HCC patients undergoing immunotherapy, we conducted an analysis on prognosis differences in high and low TRAF3IP3 expression groups among HCC patients receiving immunotherapy. Our findings align with our initial estimations, showing that high TRAF3IP3 expression is strongly correlated with favorable prognosis in tumor patients undergoing immunotherapy (Figs. 7C and 7D). Additionally, we utilized the ‘pRRophetic’ algorithm to estimate IC50 values of 138 drugs, encompassing immunotherapy, chemotherapy, and targeted therapy for HCC patients (Figs. S5–S7). Our analysis revealed that the high-expression group displayed enhanced sensitivity to lapatinib, mitomycin, temsirolimus and paclitaxel (Figs. 8A–8D, p < 0.001). Conversely, the low-expression group demonstrated increased sensitivity to bFH535 and GW441756 (Figs. 8E and 8F, p < 0.01). Ultimately, a flowchart was employed to visually illustrate the investigative pathway of TRAF3IP3 in HCC (Fig. 9).

Figure 7 Analysis of the relationship between TRAF3IP3 expression and immunotherapy in HCC.

(A and B) The IPS score for IPS-CTLA4 blocker and IPS-PD1 blocker between TRAF3IP3 high and low expression groups. (C and D) Comparison of prognosis for IPS-CTLA4 blocker and IPS-PD1 blocker between TRAF3IP3 high and low expression groups.

Figure 8 TRAF3IP3 predicted the chemotherapeutic responses.

(A–F) The signature showed that the high expression group was negatively correlated with the IC50 of the chemotherapeutic drugs (A) lapatinib, (B) paclitaxel, (C) mitomycin, and (D) temsirolimus, whereas it was positively associate with the IC50 of (E) FH535 and (F) GW441756.

Figure 9 Flow chart.

Discussion

The protein TRAF3IP3 was initially identified for its interaction with TRAF3, which activates the JNK pathway. Previous research has shown that TRAF3IP3 was highly expressed in the vascular system of breast tumors (Bhati et al., 2008), suggesting a potential role in breast cancer development and progression. Additionally, Nasarre et al. (2018) found widespread expression of TRAF3IP3 in various cancer cell lines, particularly in melanoma cells, indicating its possible involvement in tumor growth and metastasis. Furthermore, the investigators have found TRAF3IP3 was highly expressed in gliomas and played an important role in the occurrence and development of glioma (Yang et al., 2021). In contrast to these findings, the downregulation of TRAF3IP3 in HCC was associated with poor prognosis. Notably, the relationship between TRAF3IP3 and adverse outcomes in HCC patients has not been previously documented; however, the specific mechanisms underlying this relationship have not been thoroughly investigated (Pan et al., 2020). Furthermore, the researchers observed that decreased TRAF3IP3 expression correlated with poor prognosis in patients with HBV-related HCC (Ding et al., 2022). The findings have clear implications for TRAF3IP3 as biomarkers in predicting the prognosis of patients with HBV-related HCC, and they provide new research directions as well as diagnostic and treatment options for this condition. Nonetheless, there was currently a paucity of research examining the relationship between TRAF3IP3 expression and prognosis in HCC patients, as well as the related mechanisms and the evaluation of immunotherapy.

Similarly, our study yielded consistent results. Based on data from the TCGA, GTEx, and GEO databases, as well as clinical samples, we confirmed that TRAF3IP3 is expressed at lower levels in HCC tissues compared to normal liver tissues. Generally, protein makes a more direct and obvious functional influence than mRNA in organisms. In this study, we not only assessed mRNA levels of TRAF3IP3 but also confirmed down-regulation of its protein expression in HCC. Subsequent verification through IHC analysis demonstrated consistency with TRAF3IP3 expression in HCC tissues and public datasets. Importantly, high expression levels of TRAF3IP3 were associated with a more favorable prognosis in patients. Furthermore, TRAF3IP3 expression correlated with the Ishak score and tumor status, indicating its potential as an independent prognostic biomarker for patients with HCC.

Through GO and KEGG enrichment analysis of TRAF3IP3-related genes in HCC, we observed a significant enrichment in GO terms related to immune function, specifically focusing on lymphocyte-mediated immunity and positive regulation of lymphocyte activation. Furthermore, our KEGG pathway analysis highlighted an enrichment in immune-related signaling pathways such as cytokine-cytokine receptor interaction, chemokine signaling pathway, and Th1 and Th2 cell differentiation. These findings suggest that TRAF3IP3 plays a crucial role in lymphocyte activation and immune pathways, emphasizing its importance in the immune response. Recent studies have increasingly acknowledged the significance of the tumor TME, which plays a crucial role in tumor survival, progression, and metabolic activities (Hinshaw & Shevde, 2019; Kaymak et al., 2020). The immune cell infiltration within the TME has been associated with cancer patient prognosis and the effectiveness of different treatments such as chemotherapy, radiotherapy, and immunotherapy (Kennel et al., 2022; Luo et al., 2022b). Evaluating immune infiltration patterns in HCC can provide a deeper understanding of the TME and help in developing personalized immunotherapy strategies (Guo et al., 2022). The immune cell infiltration within the TME is closely linked to the progression of HCC, with certain immune cell types such as CD4+T cells, CD8+T cells, M1 macrophages, B cells, and memory T cells being associated with a favorable prognosis, while regulatory T cells, regulatory B cells, and M2 macrophages are correlated with a poor prognosis in HCC (Ren et al., 2024). In our study, we initially investigated the correlation of TRAF3IP3 with the TME using immune scores and stromal scores calculated by the ESTIMATE algorithms, which enabled the quantification of immune and stromal components in HCC. Our results demonstrated a clear positive association of TRAF3IP3 with immune scores, stromal scores, and overall estimate scores in HCC, indicating that TRAF3IP3 may serve as a key regulator of immune cells. Subsequently, we conducted a more detailed analysis to identify specific classes of immune cells associated with TRAF3IP3 expression using CIBERSORT algorithms. The results indicated a positive correlation between the expression of TRAF3IP3 and the levels of infiltrating B cells, CD4+ T cells, and CD8+ T cells, while showing a negative correlation with M0 macrophages and M2 macrophages. Previous research has shown the anti-tumor role of CD8+ T cells (Hao et al., 2021), which can be activated to become cytotoxic T lymphocytes capable of eliminating cancer cells through mechanisms like granule release and FasL-mediated apoptosis (Farhood, Najafi & Mortezaee, 2018). CD4+ T cells, on the other hand, play a supporting role in priming CD8+ T cells (Topalian et al., 2016). Several studies have demonstrated a strong correlation between CD8+ T lymphocytes and improved patient prognosis in various cancers, including HCC (Pagès et al., 2010). Among the different immune cell populations infiltrating tumors, CD8+ T cells have shown the most significant positive impact on patient survival (Bruni, Angell & Galon, 2020). Macrophages play a significant role in HCC. They are essential in chronic liver inflammation, a critical step in the initiation and progression of HCC. Tumor-associated macrophages (TAMs) are a well-known component of the tumor microenvironment, with the majority being M2-polarized macrophages (Qiu et al., 2021). Recent research by Yao et al. (2018) demonstrated that M2-polarized macrophages promote the migration and epithelial-mesenchymal transition of LIHC cells through the regulation of the TLR4/STAT3 signaling pathway (Tian et al., 2019). Our findings offer insights into how TRAF3IP3 may facilitate immune infiltration and support its potential as an immunological biomarker.

In recent years, immunotherapy has emerged as a prominent area of research for treating a wide range of malignancies. HCC often developed in the setting of virus-related chronic inflammation, immunotherapy hold promise as a viable treatment option for HCC (Xie et al., 2022). Our study discovered that the TRAF3IP3 high-expression group exhibited elevated levels of immune checkpoint expression, suggesting that TRAF3IP3 could be a reliable predictor of immune checkpoint expression in HCC. Additionally, we observed a strong correlation between TRAF3IP3 expression and chemokines, as well as chemokine receptors in HCC, indicating a potential role of TRAF3IP3 in the immune response of tumor cells to immunotherapy. IPS was introduced to evaluate immune system activity and immune cell infiltration in the tumor microenvironment, aiming to detect potential changes in patients’ response to immunotherapy. IPS is calculated based on immune-related gene expression patterns, indicating a patient’s likelihood of responding to immunotherapy (Bagaev et al., 2021). Higher IPS scores suggest a more active immune system and a better chance of positive response to immunotherapy. In this study, IPS scores were compared between patients with high and low TRAF3IP3 expression levels to assess their responsiveness to immunotherapy. The results revealed a significantly higher response to immunotherapy in the high-expression group, providing theoretical support for the application of immunotherapy in patients with high TRAF3IP3 expression. Furthermore, our evaluation of drug sensitivity showed that the high-expression group demonstrated sensitivity to a greater number of drugs among those included in immunotherapy, chemotherapy, and targeted therapy. Collectively, these findings suggest that TRAF3IP3 could be a valuable reference for designing personalized immunotherapy strategies.

Although our research has shown promising results, our study also had limitations. Firstly, our findings regarding TRAF3IP3 as a potential protective factor for patients with HCC were based on bioinformatics methods alone, lacking confirmation through animal and cell experimental validation. Secondly, our study indicated a correlation between TRAF3IP3 and the immune response, yet due to the unavailability of information on the HCC immunotherapy cohort and the limited number of patients in our center, further exploration of this relationship was constrained. Nonetheless, our study offers a foundational perspective that can guide future research endeavors.

Conclusions

In summary, our results revealed that TRAF3IP3 serves as an independent prognostic factor in patients with HCC, with its molecular mechanism may be related to immune infiltration. Based on these findings, we speculated that TRAF3IP3 has the potential to be a valuable immune-related biomarker for predicting HCC prognosis.

Supplemental Information

Supplemental Information 1 The mRNA expression of TRAF3IP3 in HCC tissues and corresponding normal tissues from an external cohort (GSE14520, n = 221).

Supplemental Information 2 Analysis of functional enrichment from an external cohort (GSE174570, n = 114).

(A) GO and (B) KEGG analyses of DEGs in samples with low and high TRAF3IP3 expression. (C) Enrichment of genes in the typical pathways by GSEA function analysis in HCC.

Supplemental Information 3 Immune infiltration analysis of TRAF3IP3.

Comparison of stroma score, immune score, and estimate score between TRAF3IP3 high and low expression groups from external cohorts (A, GSE121248, n = 107; B, GSE60502, n = 36; C, GSE76427, n = 167; D, GSE102451, n = 59).

Supplemental Information 4 The relationship between TRAF3IP3 and chemokine receptors.

Supplemental Information 5 The correlation between TRAF3IP3 expression and the efficacy of chemotherapeutic agents.

Supplemental Information 6 The correlation between TRAF3IP3 expression and the efficacy of chemotherapeutic agents from an external cohort (GSE121248, n = 107).

Supplemental Information 7 The correlation between TRAF3IP3 expression and the efficacy of chemotherapeutic agents from an external cohort (GSE121248, n = 107).

Supplemental Information 8 The mRNA expression of TRAF3IP3 in HCC tissues and corresponding normal tissues from an external cohort (GSE14520, n = 221).

Supplemental Information 9 The mRNA expression of TRAF3IP3 in HCC tissues and corresponding normal tissues from an external cohort (GSE121248, n = 107).

Supplemental Information 10 The differentially expressed genes between high- and low-expression groups.

Supplemental Information 11 GO enrichment analysis.

Supplemental Information 12 The molecular pathways linked to TRAF3IP3 in HCC were investigated using the GSEA.

Supplemental Information 13 KEGG enrichment analysis.

Supplemental Information 14 Raw data.

We extend our appreciation to the GEO database for providing open access to data, as well as to Dan Li for her assistance in gathering materials.

Additional Information and Declarations

Competing Interests

Author Contributions

Human Ethics

Data Availability

The authors declare that they have no competing interests.

Xing Wang conceived and designed the experiments, performed the experiments, analyzed the data, prepared figures and/or tables, and approved the final draft.

Xin Gao conceived and designed the experiments, performed the experiments, analyzed the data, prepared figures and/or tables, and approved the final draft.

Airu Liu performed the experiments, analyzed the data, prepared figures and/or tables, and approved the final draft.

Yan Qin performed the experiments, analyzed the data, prepared figures and/or tables, and approved the final draft.

Zhi-Yu Ni conceived and designed the experiments, authored or reviewed drafts of the article, and approved the final draft.

Xiao Lan Zhang conceived and designed the experiments, authored or reviewed drafts of the article, and approved the final draft.

The following information was supplied relating to ethical approvals (i.e., approving body and any reference numbers):

This study was approved by Research Ethics Committee of the second hospital of Hebei Medical University (2022-R706) and obeyed the Helsinki Declaration’s criteria.

The following information was supplied regarding data availability:

The data is available at NCBI GEO: GSE174570, GSE121248, GSE54236, GSE60502, GSE76427, GSE102451, GSE121248, GSE14520.

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
