# Peer review of "Study of TRAF3IP3 for prognosis and immune infiltration in hepatocellular carcinoma"

_PeerJ, doi:10.7717/peerj.18538_

## Round 0.1 · original submission · Major Revisions

Dear authors,

We kindly request that you carefully review the comments provided by the reviewers. Their valuable suggestions offer insights to enhance your manuscript. Incorporate their suggestions and carefully address all comments in your manuscript; it will significantly strengthen its content.

Reviewer 4 has suggested that you cite specific references. You are welcome to add it/them if you believe they are relevant. However, you are not required to include these citations, and if you do not include them, this will not influence my decision.

Thank you.

Reviewer 1 ·

Basic reporting

The study delves into TRAF3IP3's role in hepatocellular carcinoma (HCC), revealing elevated mRNA but reduced protein expression. High TRAF3IP3 levels align with improved prognosis, activating immune pathways, particularly B lymphocyte-mediated ones. I found the manuscript straightforward, but there were a few points that were confusing and the main conclusion is not convincing. I will elaborate on these points below.

Experimental design

1. The authors utilize various GSE datasets for validation in different analyses without explaining selection criteria. For instance, GSE14520 is exclusive to prognosis analysis, and GSE174570 solely for enrichment analysis. Both analyses should be performed in both datasets for clarity.
2. The method section should specify antibody concentration.
3. Figure 1A should highlight HCC.
4. The quantification method of TRAF3IP3 expression in FFPE stainings needs more detailed documentation.

Validity of the findings

The authors suggest TRAF3IP3 as a promising HCC prognostic biomarker. However, the data questions TRAF3IP3's clinical utility as a prognostic biomarker. Firstly, the mRNA expression difference between HCC and normal tissue is marginal. Second, higher TRAF3IP3 correlates with HCC occurrence, yet within HCC patients, high expression links to better survival. Third, while mRNA increases, protein expression decreases in HCC, creating confusion. These points collectively suggest a non-essential role for TRAF3IP3 in HCC.

Reviewer 2 ·

Basic reporting

The article conveys scientific concepts clearly but major improvement in sentence phrasing is required.

Additionally, there are several grammatical errors.

Experimental design

The experimental design is satisfactory.

Validity of the findings

1. The explanation for why TRAF3IP3 mRNA does not match protein levels is insufficient. Please provide perspective as to why that may be the case.
2. Figure 1 Please include HCC TRAF3IP3mRNA expression in the main graph.
3. Separate the C, and D headings to clearly depict the graph vs, representative images.
4. The data presented in Figure 6 is not sufficient to conclude that TRAF3IP3 is more effective than other factors in selecting HCC patients for ICI therapy as stated in line 257.
5. Discussion: Please provide more detailed insights into the existing literature and how they correlate/contradict the data to strengthen the rationale for the study.

Additional comments

The article provides sufficient evidence for TRAF3IP3 to be an independent marker in HCC, with potential clinical implications.

Reviewer 3 ·

Basic reporting

1. Basic Reporting
The title of the study clearly states the main findings of the paper: TRAF3IP3 is a biomarker with prognostic value and is connected to immune infiltration in HCC. In this, specifying whether the correlation with immune infiltration is positive or negative could be helpful.

The writing is clear and concise with literature references, making it easy to understand the research conducted. Technical terms are used correctly and consistently.

Experimental design

The authors carried out original primary research within aims and Scope of the journal utilizing the data from databases while also generating new data.
Research questions are mostly well defined, relevant & meaningful. It is stated how research fills an identified knowledge gap.

Line 106-107 “We chose 20 patients with HCC who underwent surgical excision at The Second Hospital of 107 Hebei Medical University in 2021.” What was the criteria to chose these 20 HCC patients?
Figure 1D is not properly labeled, what color shows TRAF3IP3 protein expression and how it has been quantified? The difference is not reflected in the picture.
In Line 162, Is the median value of the HCC or IPSs?
Line 191- 193 states “The immunohistochemical results of paired samples of 20 patients with HCC from The Second Hospital of Hebei Medical University also indicated the higher expression of TRAF3IP3 protein in HCC tissues than that in normal liver tissues” but Figure 1C shows higher expression in the normal samples.

Line 226-227 “The molecular processes impacted by TRAF3IP3 in HCC were further identified using GSEA.” This statement seems over-statement as there is only association and other mechanistic investigations have not been done.

Address potential limitations, such as the heterogeneity of HCC and the influence of other factors on prognosis.

Suggest future directions for research, including functional experiments and larger clinical validation studies.

Validity of the findings

Overall, the study shows promise in identifying TRAF3IP3 as a potential prognostic biomarker and highlighting its relevance to the immune response in HCC. However, the abovementioned issue exists in the current version of the study. Conclusions seem to be well stated and corrections/explanations of the study will be helpful.

Reviewer 4 ·

Basic reporting

This study presents a comprehensive analysis of TRAF3-interacting protein 3 (TRAF3IP3) and its role in hepatocellular carcinoma (HCC), utilizing an extensive range of data sources including The Cancer Genome Atlas (TCGA), Gene Expression Omnibus (GEO) cohort GSE 14520, immunohistochemical staining, and the Cancer Immunome Atlas (TCIA). The authors have done a commendable job in dissecting the complex nature of TRAF3IP3's expression in HCC, illustrating an intriguing paradox where mRNA expression levels are increased, yet protein levels are decreased in tumor samples compared to normal liver tissue.

Experimental design

However, there are some areas where the study could be further strengthened.
1. While the analysis includes a variety of datasets and employs robust statistical methods, the sample size for analysis is relatively small. Auther should expand the sample size to validate these findings further.
2. Additionally, mechanistic studies to elucidate the pathways through which TRAF3IP3 affects tumor immunity and drug sensitivity in HCC would provide valuable insights into its role as a potential therapeutic target.
3. A conclusion figure (graphical abstract) will be very useful for the readers.
4. The author could further check the key gene expression in their own tissues or cell.
5. Since this manuscript is bioinformatics-related research, some bioinformatics articles should be included in the references. The articles such as PMID: 35027051, PMID: 35413690 and PMID: 34737618 contain a lot of bioinformatics content that the author can refer to.

6. The author is advised to polish the English.
7. Data and code need to be shared either through a code-sharing repo like GitHub or a docker-like system such as codeocean for clear reproducibility of the work.

Overall, further studies exploring the mechanistic basis of these relationships is necessary for translating these findings into clinical practice.

Validity of the findings

.

---

## Round 0.2 · Minor Revisions

The authors have conducted great work on this manuscript, incorporating feedback from reviewers to enhance the clarity and depth of the content. They have successfully elucidated the role of TRAF3IP3(prognostic biomarker) and its immune infiltration impact in hepatocellular carcinoma. This manuscript now offers a clearer and more insightful understanding of the mechanisms at play, thanks to the thoughtful integration of the reviewers' suggestions.

Comments :

The figure legend is incorrect in the case of Figure 1, please check. D should be changed to E and vice versa. "(D), The quantitative analysis of TRAF3IP3 expression in 20 paired HCC tissue based on average optical density. (E) Representative IHC staining of TRAF3IP3 in HCC tissues and adjacent normal tissues. (*p < 0.05; **p < 0.01; ***p < 0.001). "

1. "Citations are missing. Please evaluate how this study is different or furthers the field:
This study from 2022 found both TRAF3IP3 and SLAMF1 as highly expressed in tumor tissues:
Ding W, Zhang Z, Ye N, Zhuang L, Yuan Z, Xue W, Tan Y, Xu X. Identification of Key Genes in the HBV-Related HCC Immune Microenvironment Using Integrated Bioinformatics Analysis. J Oncol. 2022 Oct 15;2022:2797033.

2. This study found that high TRAF3IP3 levels predict Poor prognosis of glioma patients:
https://www.sciencedirect.com/science/article/abs/pii/S1878875021000255

3. Others:
https://www.wjgnet.com/1007-9327/full/v26/i8/789.htm
Please do a thorough literature search, it appears that this is not a new finding."

4. The authors should again review the literature, add all relevant citations, and explain how what they did is different and how their study will further the field.

5. TRAF3IP3 is in the title of the paper, and in fact, it says that is a prognostic marker. If they did not validate it as a marker, the title is misleading. Authors should addresses these comments.

Thank you

Reviewer 2 ·

Basic reporting

The manuscript has improved significantly.

Experimental design

The methodology section of the article is comprehensive and well-structured.

Validity of the findings

The results presented in this article summarize the key findings

Reviewer 3 ·

Basic reporting

The manuscript is unambiguous, professional English is used throughout. Literature references, and sufficient field background/context is provided.

The figure legend is incorrect in the case of Figure 1, please check. D should be changed to E and vice versa.  "(D), The quantitative analysis of TRAF3IP3 expression in 20 paired HCC tissue based on average optical density. (E) Representative IHC staining of TRAF3IP3 in HCC tissues and adjacent normal tissues. (*p < 0.05; **p < 0.01; ***p < 0.001). "

Experimental design

The manuscript contains original primary research within the Aims and Scope of the journal. The authors have significantly improved the manuscript by addressing the reviewers' comments.

Validity of the findings

The authors have addressed the reviewers’ comments by incorporating improvements. The added clarity and detailed explanations enhance the manuscript. The manuscript is now of publishable quality.

---

## Round 0.3 · Minor Revisions

Dear authors,

We kindly request that you carefully review the final comments provided by the reviewer. They suggested few valuable suggestions , it will enhance your manuscript. Incorporate their suggestions and carefully address all comments in your manuscript. Thanks

Reviewer 3 ·

Basic reporting

The research article has made improvements based on the reviewer's feedback but the discussion part has deteriorated in some areas (refer to line 304-312)
This paragraph, while generally understandable, has some minor issues in terms of English usage and style that could be improved:
"In the meantime": This phrase feels a bit out of place and doesn't connect to the previous or following sentences. It could be replaced with a more specific transitional phrase like "Similarly" or "Consistent with previous findings."
Several sentences start with "Our analysis..." or similar phrases. This creates monotony.
Bioinformatics analysis? Refer to methods and datasets so that it's more clear.

Experimental design

The study design is clear and matches with primary research within Aims and Scope of the journal.

Research question well defined, relevant & meaningful.

Validity of the findings

Conclusions are well stated, linked to original research question & limited to supporting results.

---

## Round 0.4 · accepted · Accept

The manuscript is now acceptable to move forward in the approval process.